# Effect of Drug Encapsulation and Hydrothermal Exposure on the Structure and Molecular Dynamics of the Binary System Poly(3-hydroxybutyrate)-chitosan

**DOI:** 10.3390/polym15102260

**Published:** 2023-05-10

**Authors:** S. G. Karpova, A. A. Olkhov, I. A. Varyan, A. A. Popov, A. L. Iordanskii

**Affiliations:** 1Department of Biological and Chemical Physics of Polymers, Emanuel Institute of Biochemical Physics, Russian Academy of Sciences, 4 Kosygina Street, 119334 Moscow, Russia; karpova@sky.chph.ras.ru (S.G.K.); anatoly.popov@mail.ru (A.A.P.); 2Academic Department of Innovational Materials and Technologies Chemistry, Plekhanov Russian University of Economics, 36 Stremyanny Lane, 117997 Moscow, Russia; 3N. N. Semenov Federal Research Center for Chemical Physics Academy of Science, 119991 Moscow, Russia

**Keywords:** poly-(3-hydroxybutyrate), TEMPO, stable radical, correlation times, amorphous phase

## Abstract

In this work, film materials based on binary compositions of poly-(3-hydroxybutyrate) (PHB) and chitosan with different ratios of polymer components in the range from 0/100 to 100/0 wt. % were studied. Using a combination of thermal (DSC) and relaxation (EPR) measurements, the influence of the encapsulation temperature of the drug substance (DS) of dipyridamole (DPD) and moderately hot water (at 70 °C) on the characteristics of the PHB crystal structure and the diffusion rotational mobility of the stable TEMPO radical in the amorphous regions of the PHB/chitosan compositions is shown. The low-temperature extended maximum on the DSC endotherms made it possible to obtain additional information about the state of the chitosan hydrogen bond network. This allowed us to determine the enthalpies of thermal destruction of these bonds. In addition, it is shown that when PHB and chitosan are mixed, significant changes are observed in the degree of crystallinity of PHB, degree of destruction of hydrogen bonds in chitosan, segmental mobility, sorption capacity of the radical, and the activation energy of rotational diffusion in the amorphous regions of the PHB/chitosan composition. The characteristic point of polymer compositions was found to correspond to the ratio of the components of the mixture 50/50%, for which the inversion transition of PHB from dispersed material to dispersion medium is assumed. Encapsulation of DPD in the composition leads to higher crystallinity and to a decrease in the enthalpy of hydrogen bond breaking, and it also slows down segmental mobility. Exposure to an aqueous medium at 70 °C is also accompanied by sharp changes in the concentration of hydrogen bonds in chitosan, the degree of PHB crystallinity, and molecular dynamics. The conducted research made it possible for the first time to conduct a comprehensive analysis of the mechanism of action of a number of aggressive external factors (such as temperature, water, and the introduced additive in the form of a drug) on the structural and dynamic characteristics of the PHB/chitosan film material at the molecular level. These film materials have the potential to serve as a therapeutic system for controlled drug delivery.

## 1. Introduction

Currently, a significant amount of scientific and practical developments is devoted to the use of biodegradable composites/blends in biomedicine, the packaging industry, and tailoring environmental problems [1,2,3]. In the framework of the circular economy [4,5,6], for biomedicine platforms and food packaging, there is a direct tendency to move from conventional synthetic plastics to sustainable and biodegradable ones [5,6]. One of the key reasons for the intensive use of such materials is the fact that polymers’ blending could lead to not only a significant improvement in performance, but also the appearance of new physicochemical parameters of the composites that were not inherent in the original components. The triad made of biodegradable polymers such as poly(3-hydroxybutyrate) (PHB), polylactide (PLA), and chitosan has practically inexhaustible sources of (bio)chemical synthesis and covers a wide range of hydrophilicity, and its constituents are characterized by high ecological and cell/tissue compatibility. In addition, these bio-based polymers are often used in combination with other polymers as constructional materials, innovative active packaging, and materials for environmental remediation [7,8,9].

The wide range of implementations enables the above biopolymers to dominate the world trade markets of bio-materials and bio-articles [10,11,12].

PHB is the main representative of natural polyesters of the polyhydroxyalkanoate family [11,12,13]. Along with useful properties, this polyester has a number of undesirable characteristics: high cost and fragility. To overcome these drawbacks, its copolymers with the units of the PHA family such as 3-hydroxyvalerate, 3-hydroxyoctanoate, and 4-hydroxybutyrate are used [12,13,14]. The second way to improve PHB characteristics for the exploitation object is the development of blends and composites that modify its behavior in the implementation areas from biomedicine to environmental safety [15,16].

Chitosan is the second component of the studied binary bio-based system, and it is also biocompatible and capable of biodegradation. Similar to PHB, it is widely used in biomedical, packaging, and eco-friendly applications such as products of tissue engineering, drug delivery therapeutic systems, active barriers with inherent functionality, and adsorbents for environment remediation [17,18,19,20]. Unlike PHB, it is characterized by high hydrophilicity, which is determined by the presence of a large number of functional groups, such as amine and hydroxyl groups. Amine groups form strong intramolecular and intermolecular hydrogen bonds, which transform chitosan into a glassy state. This material is able to be resorbed in the environment into environmentally friendly degradation products for a sufficiently long time—from a month to a year. An important feature of both polymers, as well as the products made from them, is the biodegradation of the macromolecular chains by the mechanism of hydrolytic or enzymatic destruction. The end products of the decomposition process are products that are safe for the body: for PHB, these are carbon dioxide and water. However, the high sensitivity of chitosan to moisture limits its application. This disadvantage can be overcome by mixing chitosan with moisture-resistant polymers such as PHB [21,22,23], polylactides [23], and polycaprolactone [24] provided that the biodegradability of the blends’ constituents must be saved.

The bio-based polymers’ disintegration mechanisms, namely biodegradation, oxidation, and hydrolysis, are significantly affected by the structural organization of the amorphous and crystalline phases formed in the bulk of the composite [25,26,27]. The rate of these processes increases if the structure of the sample is not homogeneous, but partly or fully heterogeneous. The semicrystalline biopolymers in the blends have a mutual effect on their crystallization behavior and glass transition, leading to a change in diffusion-transport characteristics, drug delivery, kinetic biodegradation, oxidation/ozonolysis, hydrolysis, and thermal stability of composite films and fibers [28,29,30].

At mixing, hydrophobic PHB and extremely hydrophilic chitosan, which are located at the opposite ends of the polarity scale, can generate composites in a wide range of hydrophilicity. By varying the composition of the PHB/chitosan blends and thereby influencing their morphology and crystallinity, it is possible to obtain versatile materials with different physical and chemical characteristics and accordingly diverse functionalities, such as water diffusivity, appropriate thermal and mechanical behavior, and controlled degradation rate [9,21,31,32].

An effective way to assess the state of the amorphous and crystalline phases of both initial biopolymers (PHB and chitosan) and their blends is a combination of dynamic and structural methods. In this work, EPR spectroscopy (probe analysis) and differential scanning calorimetry (DSC) were used. Such a combination of research methods allowed the authors to obtain a more complete assessment of the structural evolution of a PHB/chitosan blend in an aqueous medium in the short time interval (minutes and hours), which precedes the hydrolytic decomposition of the polymer system.

The main purpose of this work was to study the effect of the composition of PHB/chitosan blends and the encapsulation of the drug substance (dipridamole) on the crystal structure, the state of hydrogen bonds of polymer molecules, and the segmental mobility of the polymers. In addition, the EPR method was used to study the influence of external aggressive factors, temperature, and the aqueous medium on the process of breaking the hydrogen bond network, as well as the evolution of PHB crystallinity and the molecular dynamics of a microprobe (TEMPO) in composites. The presented study will allow a deeper understanding of the effect of the loaded drug on the state of the composite system as well as contribute to the improvement of existing and the development of new therapeutic platforms and scaffolds with the function of controlled drug release.

## 2. Materials and Methods

In the work, mixed compositions based on biodegradable polymers hydroxybutyrate and chitosan were studied. To obtain films, we used a natural biodegradable polymer poly(3-hydroxybutyrate) (PHB) of the 16F series (BIOMER, Krailling, Germany), obtained by microbiological synthesis. The original polymer was a white fine powder. The molecular weight of PHB was M_w_ = 2.06 ⋅ 10^5^ g/mol (206 kDa), density d = 1.248 g/cm^3^, T_m_ = 177 °C. Chitosan (Bioprogress, Shchelkovo, Russia), an infusible polysaccharide, was a fine powder. The molecular weight of this polymer was M_w_ = 4.4 ⋅ 10^5^ g/mol; degree of deacetylation—82.3%. Solvents were used in the production of films by casting: for PHB—CHCl_3_ and dioxane grade ChDA (ZAO Ecos1, Russia); for chitosan—CH_3_COOH grade ChDA. Films were prepared by mixing a solution of chitosan in aqueous acid media and a solution of PHB in dioxane. Chitosan solution was obtained by dissolving the powder in acetic acid. Dipyridamole was used as a model drug for controlled release (2,2′,2″,2‴-((4,8- Di-1-piperidinylpyrimido (5,4-d)pyrimidine- 2,6-diyl)dinitrilo]tetrakis(ethanol)). The pharmacological group to which dipyridamole (DPD) belongs is antiplatelet agents, angioprotectors and microcirculation correctors, and adenosinergic agents. DPD is a yellow crystalline powder. The molecular weight of DPD is 504.53 g/mol. Ultrathin PHB fibers were obtained by electroforming (ES). To obtain fibers, molding solutions were prepared: PHB and PHB with DPD in chloroform. The concentration of PHB in the solution was 7% by weight. The content of dipyridamole in the polymer composition was 5% by weight, relative to the mass of PHB. PHB molding solutions with DPD were prepared using an automatic magnetic stirrer with heating and an ultrasonic bath. The following solvents were used to obtain films by casting: for PHB—CHCl_3_ and dioxane of the ChDA brand (JSC Ekos-1, Moscow, Russia); for chitosan—CH_3_COOH of the ChDA brand. Chitosan solution was obtained by dissolving the powder in acetic acid. Films were prepared by mixing a solution of chitosan in aqueous acid media and a solution of PHB in dioxane.

X-band electron paramagnetic resonance (EPR) spectra were recorded on an EPR-V automated spectrometer (Federal Research Center for Chemical Physics, Russian Academy of Sciences, Moscow, Russia). The value of the microwave power to avoid saturation effects did not exceed 1 mW. The modulation amplitude was always significantly less than the width of the resonance line and did not exceed 0.5 G. The stable nitroxide radical TEMPO was used as a spin probe. The radical was introduced into the fibers from the gas phase at a temperature of 60 °C for an hour. The concentration of the radical in the polymer was determined by integrating the EPR spectra. The reference was an evacuated TEMPO solution in CCl_4_ with a radical concentration of ~1·10^−3^ mol/L.

The probe rotation correlation time τ was found from the EPR spectra using the formula given earlier [33]:τ = ΔH^+^ × [(I^+^/I^−^)^0.5^ − 1] × 6.65 × 10^−10^(1)
where ΔH^+^—width of the spectrum component located in a weak field, I^+^/I^−^—intensity ratio of the components in a weak and strong field. The measurement error for τ was ± 5%.

The equilibrium concentration of the radical adsorbed in samples of the studied compositions of the same mass was calculated using Brucker (winer) software. In the process of identifying the spectra, the identified, weighed samples were recorded, so the calculation of the radicality in each sample was performed using the Origin program. The weight of the polymer was 0.02–0.03 g.

The samples were studied by DSC using a DSC Q-20 instrument from TA Instruments (New Castle, DE, USA) in a nitrogen atmosphere at a heating rate of 10 K/min. The average statistical error in measuring thermal effects was ±3%.

Water and thermal impacts on the polymer were determined as follows: Samples were placed in a bottle with distilled water and then placed in an oven and kept at T = 70 °C for 2 h. Then these samples were dried for 7 days, and their weight was checked. After 5 days, the weight of the samples did not change.

## 3. Results and Discussion

### 3.1. Thermal Characteristics of PHB/Chitosan Compositions with Encapsulated DPD Exposed in an Aqueous Medium

#### 3.1.1. Binary Composition of PHB/Chitosan

When mixing two polymer components, PHB and chitosan, one should expect changes in their thermal, mechanical, and diffusion characteristics along with variability in the crystallinity and morphology of the mixture composition. In this work, the DSC method was used to study the crystalline phase of PHB in its mixed compositions with chitosan. For example, Figure 1 shows the heating endotherms of the composition 40/60% (PHB/chitosan) for binary (a) and triple compositions of PHB/chitosan/DPD (b) at 5% wt. The DSC curves of polymer samples of various compositions with component ratios from 80/20 to 30/70% showed the presence of three characteristic endothermic peaks in all thermograms.

With a rise in temperature, e.g., under heating of the chitosan samples, as provided by the DSC protocol, the partial and then full disrupture of the cross-linking formed by their hydrogen bonds transpires. On the thermograms, this process is accompanied by an endothermic gently sloping maximum that characterizes the intensity of H-bond destruction [23,34,35]. The second and third peaks correspond to different modes of PHB melting. The bimodal nature of the melting of PHB manifests itself in a wide range of compositions (see Table 1). The exception is samples with extremely low PHB/chitosan ratios, namely 0/100 and 20/80%, for which, as a result of extremely low crystallinity, melting endotherms are not observed. The melting region of the PHB homopolymer (Tm) has two maxima at 157 and 173 °C, which indicates the existence of two populations of the biopolyester crystal structure, differing both in size and degree of their perfection with a total specific enthalpy of melting ∆H = 74.4 J/g and a degree of crystallinity χ = 52% (see Table 1). The high-temperature peak in the region of 173 °C belongs to the melting of crystals with a more perfect structure, and the low-temperature peak in the region of 157 °C belongs to the melting of a less perfect PHB crystal structure. The bimodal nature of the phase transition of PHB was observed earlier for its ultrathin fibers and films [36,37]. With an increase in the content of chitosan, a slight shift in the T_m_ values to the low-temperature region is due to the influence of its molecules on the process of biopolyester crystallization.

Figure 2 and Table 1 present data on PHB crystallinity (χ), enthalpy (∆H), and hydrogen bond breaking temperature (T_D_) of chitosan, as well as melting temperatures of the PHB crystalline fraction (T_m_) on the composition of the system. As noted above, in mixtures with a low concentration of biopolyester (<30%), its crystalline phase is not formed; therefore, all curves reflecting the thermal characteristics of PHB do not start from the zero composition of the mixture, but from the ratio of 30/70% (see Figure 2a). With this composition, the crystallinity is ~33%, which is almost 2 times lower than the crystallinity of the PHB homopolymer, and then, as its content in the system increases, the values of χ increase rapidly. The “crystallinity–composition” curve includes a break in the range of compositions at 50/50% PHB/chitosan, and with a higher content of PHB in the mixture, the crystallinity of PHB changes monotonously and rather weakly. In the opposite way, the enthalpy and cleavage temperature of H-bonds increase as the concentration of chitosan increases. The characteristic point of compositions (50/50%) appears most clearly for curve 1 in Figure 2b. It is also clearly seen here that in the region of high PHB concentrations (>50%), the dependence of ∆H on the composition is extremely weak. Similarly, the position of the flat maximum of the DSC curve, which belongs to chitosan and characterizes the temperature of the destruction of its hydrogen bonds (T_D_), shifts from 126 to 85 °C. The DSC curves also show that the destruction of hydrogen bonds in native chitosan occurs at 150–104 °C. In mixed compositions, this process is realized at lower temperatures: 130–50 °C. The enthalpy of destruction of these bonds for chitosan in PHB/chitosan mixtures occupies a wide range of values. At a lower concentration of chitosan in the mixture (<30%), the maximum belonging to the heat of rupture of hydrogen bonds in the system was not observed.

A decrease in the degree of crystallinity and a decrease in the melting point of PHB with an increase in the content of chitosan indicate that the crystalline phase of PHB in the binary mixture under study becomes less perfect. In the process of crystallization, polysaccharide molecules exert a steric hindrance to the perfect packing of polyester molecules in crystallites and lamellae to a greater extent, the higher the content of polysaccharide.

The nature of the change in the thermal characteristics of the binary PHB/chitosan system depending on the composition of the polymer components allows us to make a preliminary conclusion that a specific concentration region is observed, 50/50%, the transition through which is accompanied by phase inversion, i.e., when the continuous (dispersed) phase of chitosan is transformed into dispersed. If a strong and dense network of hydrogen bonds is formed in the initial chitosan, which retains water molecules well, then as PHB molecules are introduced, a sharp drop in ∆H occurs, reflecting the reduction of such bonds and, consequently, a rapid decrease in the integral heat of their rupture. On the contrary, the increase in the crystallinity of the second component, PHB, shows that its macromolecules encounter fewer and fewer obstacles in the formation of crystalline regions. Along with steric limitations of crystallization, the role of intermolecular hydrogen bonds formed during the interaction between chitosan molecules containing amine groups and PHB molecules, including both ester groups of the main chain and terminal acid and hydroxyl fragments, cannot be excluded. Although such an interaction seems to be weaker than the energy of chitosan hydrogen bonds, it can also lead to the decompression of the network of chitosan hydrogen bonds, i.e., to an increase in the free volume in the system. The latter effect should enhance the segmental mobility of polymers, which will be analyzed further using the EPR method.

#### 3.1.2. The Ternary System of PHB/Chitosan/Drug

In the transition from the binary to the ternary composition of PHB/chitosan/DPD, let us consider the effect of the encapsulated drug on the crystal structure of PHB and the strength of the hydrogen bonds of chitosan. In therapeutic systems and active packaging, the impermeable crystalline phase creates steric hindrances to DPD diffusion transport. Therefore, its introduction into the system is of interest both for the analysis of the structural evolution of a mixture as a matrix that provides control of drug delivery, and for the assessment of segmental mobility, which determines such key processes in polymers as diffusion, barrier properties, sorption capacity, and controlled release.

Figure 2a (curve 2) shows the dependences of the degree of crystallinity of PHB in the ternary system PHB/chitosan/DPD. A comparison of these curves with similar results for the binary system shows that for all compositions of the ternary system, the degree of crystallinity of PHB with encapsulated DPD is higher than that in the initial binary systems. Similar to the binary system, the enthalpy of hydrogen bond breaking in the ternary system tends to decrease, and most importantly, they both intersect in the composition range of approximately 50/50%, i.e., at the same inversion point (see Figure 2b). The excess of crystallinity in the presence of DPD can be explained either by the additional nucleation of the crystalline phase, where the low-molecular-weight drug acts as a nucleation agent, or by the already mentioned plasticizing effect of DPD. In the range of compositions above the inversion point, the concentration of hydrogen bonds in the ternary system becomes noticeably lower than that in the binary system and also changes slightly with increasing PHB content. The data presented earlier in the work [38] for ultrathin PHB fibers also showed that the encapsulated drug in ultrathin PHB fibers affects the crystallization that occurs during electrospinning.

#### 3.1.3. Hydrated Binary System PHB/Chitosan

At the next stage of the analysis of the properties of the PHB-chitosan system, we will consider the effect of water exposure on its structure at a standard elevated temperature of 70 °C. It is well known that when polymers come into contact with water, their structure and crystallinity change [39,40], and during water evaporation (drying), the polymer crystallinity is increased [41]. In each individual case, it is necessary to establish the degree of structural changes. It is especially important to do this for an amphiphilic blended composition formed by hydrophobic PHB and hydrophilic chitosan, which are often used in humid conditions in combination with elevated temperatures.

In general, water molecules can interact with polymer molecules in two different ways. On the one hand, they have a plasticizing effect on hydrophilic and moderately hydrophilic polymers, which, as a result of an increase in segmental mobility, can lead to an additional increase in the crystalline phase, i.e., to their recrystallization [42]. On the other hand, after desorption of water from the polymer matrix, water complexes remain in the form of clusters, which are typically characteristic of hydrophobic polymers, in particular in Zimm’s clustering integral modeling [43,44], or hydrated functional groups, which is more typical for polysaccharides, proteins, polyalcohols, and other hydrophilic polymers [45].

The data on the degree of crystallinity of samples of the PHB/chitosan binary mixture exposed to distilled water at 70 °C for 120 min are shown in Figure 2a (curve 3). It can be seen from the figure that, after exposure to an aqueous medium, PHB/chitosan films with a composition of <50/50% crystallinity have higher values compared to the initial samples. This indicates the predominance of the plasticizing effect of water and, as a result, pre-crystallization in the participation of maximally oriented PHB feed-through circuits. Beyond the inversion point, i.e., in the range of values of the composition >50/50%, when its continuous phase is formed due to the hydrophobization of the system, the sorption of water decreases and the plasticizing effect is less noticeable. As a result, the increase in crystallinity is quite small.

Chitosan is a hydrophilic polymer. Exposure to an aqueous medium leads to the hydration of its functional groups and then leads to swelling and, as a result, to a sharp increase in the free volume of the polysaccharide [46]. A comparison of curve 3 and the previously considered curves 1 and 2 (see Figure 2b) shows that as a result of exposure to water of mixtures of composition <50/50%, the enthalpy of hydrogen bond rupture noticeably decreases. Additionally, the destruction of the hydrogen bond network can be indicated by a drop in the temperature of hydrogen bond destruction, shown in Figure 2c. A characteristic feature of all three curves in Figure 2b is the preservation of the inversion point at ~50% ratio of polymer components; they all intersect with each other near this point.

#### 3.1.4. Hydrated Ternary System PHB/Chitosan/DPD

In the conclusion of this section, let us consider the simultaneous effect of encapsulated DPD and exposure in an aqueous medium on the thermal characteristics of the initial binary composition of PHB/chitosan. Curves 4 on the corresponding fragments a, b, and c in Figure 2 reflect this effect. The crystal structure for this system, as well as for the non-aqueous ternary PHB/chitosan/DPD system (curve 3), begins to appear somewhat earlier on the concentration scale, namely already in samples containing 20% PHB containing 15–17% of the crystalline phase. A further sharp increase in the values of χ from ~15% to ~65% demonstrates the key effect of the drug and plasticizer on the crystallinity of PHB.

Up to the inversion point, i.e., at a high content of chitosan, its contact with water leads to intense swelling and an increase in the segmental mobility of polysaccharide chains. At the same time, the plasticization of the polyester, PHB, appears in an aqueous medium at a moderately high temperature. Both of these effects affect the crystallization ability of the latter in different ways. If chitosan in the absence of a plasticizer (water), as noted above, prevents the packing of PHB chains into crystallites and lamellae, then an increase in its segmental dynamics upon contact with hot water leads to (a) an increase in the crystallinity of the biopolyester and (b) an expansion of the range of PHB concentrations, where crystallization occurs. If the formation of a crystalline phase is observed only at a composition of 30/70% or more for non-watered systems PHB/chitosan and PHB/chitosan/DPD, then in similar hydrated systems, the presence of crystals is recorded at a lower composition value (at 20/80%). If we compare the crystallinity of hydrated systems that do not contain and contain DPD, curves 3 and 4 in Figure 2a, then the increase in crystallinity is more than 240%, namely 15% and 37%, respectively. Based on this difference, it can be concluded that, other things being equal, the encapsulation of DPD in a hydrated polymer medium does not contribute to the pre-crystallization of PHB. A similar negative effect of the presence of DPD is also observed when comparing the thermal effects of hydrogen bond breaking in chitosan (Figure 2b). Indeed, up to the inversion point, the hydrated PHB/chitosan composition (curve 3) has a stronger formation of hydrogen bonds than the same PHB/chitosan/DPD composition (curve 4). In general, the treatment of samples of mixed compositions with water at 70 °C has a stronger effect on their structural and thermal properties than the introduction of drugs. This is clearly seen when comparing the location of curves 1 and 2 in Figure 2b for the original systems with the location of similar curves 3 and 4 for watered systems. With this comparison, the difference in ΔH values is very clear, which shows the great role of water sorption as a specific aggressive agent.

### 3.2. Dynamic Characteristics of the Amorphous Phase of Mixed Compositions of PHB/Chitosan with Encapsulated DPD after Exposure to an Aqueous Medium

In partially crystalline polymers, the structure of amorphous regions is largely determined by the influence of their crystalline phase. Therefore, mixing highly crystalline PHB and practically non-crystallizing chitosan changes not only the degree of crystallinity of the biopolyester, but also the molecular dynamics in the amorphous regions formed mainly by the polysaccharide. To study molecular/segmental mobility, the EPR method was used using the stable TEMPO nitroxide radical, which acts as a molecular probe.

Let us consider the effect of the composition of the PHB/chitosan mixture on the dynamics of polymer molecules. Previously, it was shown that the EPR spectra of the radical in the matrices of the PHB homopolymer and the PHB/chitosan binary system represent a superposition of two spectra corresponding to two populations of radicals with their own characteristic correlation times τ_1_ and τ_2_ [47,48].

Time τ_1_ reflects molecular mobility in denser amorphous regions with low free volume (slow component of the spectrum), while τ_2_ reflects mobility in less dense regions with higher free volume (fast component of the same spectrum). The transition of the speed of rotation on the TEMPO radical from fast to slow is shown in Figure 3. At the same time, the heterogeneous nature of amorphous regions is due to the difference in the packing density of polymer molecules and, consequently, the difference in molecular dynamics. However, as shown by Chumakova [49], the characteristic mobility in a certain range of probe rotation speeds can be represented by a single integral value—the characteristic relaxation time (τ).

Because of their high potential barrier to internal rotation of chain links, polysaccharide molecules such as chitosan are highly rigid. This potential (energy barrier) is due to steric hindrances in the movement of the elementary link, as well as the formation of a network of hydrogen bonds. Depending on the history of the chitosan sample, this network may include immobilized water molecules. The glassy dense network of this polysaccharide prevents the effective penetration of the radical, which is confirmed by its low concentration in samples with a low PHB/chitosan ratio (up to 30/70%), as shown in Figure 4a. With a further increase in the concentration of biopolyester in the mixture, the concentration of the radical increases sharply and reaches a maximum in the region of ~60% PHB. As already shown in Section 3.1, the range of 50–60% belongs to the inversion point, where practically all thermal characteristics change the nature of the dependence on the composition. Due to the low compatibility of the components and the phase transition, the dispersed phase of PHB passes into the dispersion phase. In this case, the loosest structure is formed. At a high content of PHB (~50–100%), the concentration of the sorbed probe also significantly decreases due to the growth of crystallinity in the polymer mixture, since the radical practically does not penetrate into the densely packed crystalline regions of PHB. Figure 4b shows the dependence of the characteristic correlation time τ on the composition of the mixture, which is symbatic with the previous dependence 4a. A maximum is also observed here, located in the same concentration range (~50% PHB). The closeness of the shape of the concentration curves, reflecting the structural (a) and dynamic (b) nature of the polymer mixture, as in the previous Section 3.1, is determined by its phase transformation, namely the transition of PHB from dispersed material to dispersion medium.

For chitosan, in addition to intra- and intermolecular hydrogen bonds, when mixed with PHB, another type of interaction arises, which is determined by the formation of hydrogen bonds between the ester groups of PHB and the amine groups of chitosan formed as a result of deacetylation. It is reasonable to assume that the introduction of PHB molecules into the composition and their interaction with chitosan molecules can partially destroy the previously formed hydrogen network of the polysaccharide, which becomes less compact as a result. This fact is confirmed by the course of the descending branch of the concentration dependence of the integral correlation time (τ) (see curve 1 in Figure 4b). The intensive growth of this characteristic in the range of low and moderate PHB concentrations (20–50%) up to the concentration inversion point of 50/50% is determined by the screening effect of polyester molecules, which hinder the contact of chitosan molecules with water. Hydrophobic PHB molecules partially prevent the penetration of water into chitosan and prevent the plasticizing and loosening effects that usually accompany the sorption of solvent molecules by chitosan samples.

Let us consider the effect of DPD on the molecular dynamics of PHB/chitosan compositions (see curve 2 in Figure 4a). It can be seen that the nature of this dependence differs little from the analogous dependence for the PHB/chitosan system, which does not contain drugs. However, in the range of high biopolyester concentrations (>50%), DPD encapsulation in the system leads to a slight decrease in the sorption capacity of the stable radical and an insignificant drop in molecular mobility.

It is well known that water, penetrating into a binary polymer matrix, can affect both its physicochemical and mechanical properties and the diffusion–sorption characteristics of the third component (drug, electrolyte, modifying agent, etc.). As previously shown [21], for monolithic film samples, mixing moderately hydrophobic PHB polymer with highly hydrophilic chitosan at room and physiological temperatures (25 and 37 °C, respectively) leads to partial hydrophilization of the system, which manifests itself in an increase in its sorption capacity and in a sharp increase in drug diffusion coefficients.

Figure 5 shows the dependences of the correlation time on the composition of the PHB/chitosan mixture for different exposure times in the aquatic environment. All curves, except for the original curve 1, have the character of a curve with an extremum and have fairly close values. Consequently, structural changes that determine the segmental mobility of the composition occur already in the first 30 min of exposure. Further heating in water to a temperature of 70 °C has little effect on the dynamics of polymer molecules. The encapsulation of the drug in the polymer system slightly shifts the inversion point corresponding to the maxima on the curves to the 60/40% PHB/chitosan composition. Here it can also be assumed that the appearance of DPD in the system contributes to the preliminary crystallization of PHB; therefore, a comparison of Figure 5a,b shows a small contribution of the drug to the dynamics of the molecular chains of the composition.

Previously, it was suggested that the transition of the phase state of PHB from dispersed material to dispersion medium shields finely dispersed regions of chitosan from the action of water molecules. With this in mind, the overall picture of the dynamic behavior of molecules is as follows: (a) Ascending branches of curves 1–4 in Figure 5, which reflect a slowdown in segmental dynamics, are due to the binding of water by the functional groups of chitosan. In the same range of compositions, a noticeable pre-crystallization of PHB is observed, which prevails over the plasticizing effect. (b) After passing the maximum at the inversion point of the compositions, the segmental mobility increases; since here the biopolyester is not shielded by chitosan molecules, it forms a dispersed phase and, in direct contact with water, undergoes its plasticizing effect with a rather weak pre-crystallization (see Figure 2a).

It should be noted that the values of τ noticeably decrease during the exposure time in the aquatic environment, which is very likely due to the diffusion of water into the volume of PHB [50]. Encapsulation of DPD in the PHB/chitosan binary system in combination with the thermal process of water exposure (Figure 5b) reduces the time of its exposure to biopolyester molecules. If, in the absence of a drug, the change in segmental dynamics occurred within 120 min and possibly a little longer, then in its presence, the process of changing the dynamics is completed much faster, in 20 min. As seen in this figure, curves 1 and 2 diverge significantly, while subsequent curves 3 and 4 practically coincide with curve 2. As already reported, the drug distributed in the polymer matrix makes the polymer structure more disordered and, due to the presence of functional groups, creates additional osmotic forces. Both of these effects increase the diffusion flow of water in the system and shorten the time for structural and dynamic changes.

Additional information on the dynamic behavior of the PHB/chitosan system of various compositions was obtained by studying the temperature dependence of the molecular probe rotation speed and determining the corresponding effective activation energy Eτ values (see Figure 6). A characteristic feature of Eτ as a function of the content of chitosan in PHB is the presence of its maximum in the region of the inversion point, which is in accordance with all previous thermal and dynamic results.

## 4. Conclusions

The study of the structural and dynamic state of the composite made it possible for the first time to analyze the effect of external factors, such as temperature, drug encapsulation, and an aqueous medium at an elevated temperature (70 °C). A decrease in crystallinity and melting temperature of PHB with an increase in the content of chitosan in the binary mixture indicates that chitosan molecules prevent the formation of the PHB crystalline phase, making it less perfect. During the crystallization of the biopolyester, the polysaccharide molecules sterically hinder the perfect packing of PHB molecules in crystallites. The change in the thermal characteristics of the PHB/chitosan system depending on its composition made it possible to identify a specific concentration region, with a composition of 50/50%, where phase inversion occurs, namely the process where the continuous (dispersion medium) phase of chitosan is converted into a dispersed material. As PHB molecules are introduced into the system, the values of ∆H decrease sharply, which reflects a decrease in the number of H-bonds. The transition from the PHB/chitosan binary system to the PHB/chitosan/drug ternary system showed that the degree of PHB crystallinity increases due to additional nucleation of the crystalline phase. As a result of exposure to water at 70 °C for the compositions with a ratio of components <50/50%, the enthalpy of hydrogen bond breaking sharply decreases. Additionally, the destruction of the network of hydrogen bonds may be indicated by a drop in the temperature of destruction of hydrogen bonds. All characteristics’ dependences, namely χ, ∆H, and T_D_, for hydrated and non-hydrated compositions intersect each other near a 50/50% concentration point for both binary and ternary systems. Exposure to an aqueous medium at 70 °C is also accompanied by sharp changes in the concentration of hydrogen bonds in chitosan and molecular dynamics. The results presented herein could serve potentially for the development of drug delivery therapeutic systems.

In this work, crystallinity measurements were performed by DSC using a DSC Q-20 instrument from TA Instruments (New Castle, DE, USA) in a nitrogen atmosphere at a heating rate of 10 K/min. The average statistical error in measuring thermal effects was ±3%. (Center for Shared Use of the Institute of Biochemical Physics ”New Materials and Technologies”, Russian Academy of Sciences).

## Figures and Tables

**Figure 1 polymers-15-02260-f001:**
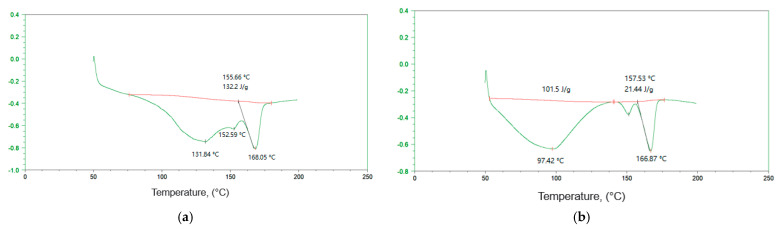
Heating endotherms of 40/60% PHB/chitosan mixed compositions: (**a**) initial binary composition and (**b**) PHB/chitosan/DPD ternary composition.

**Figure 2 polymers-15-02260-f002:**
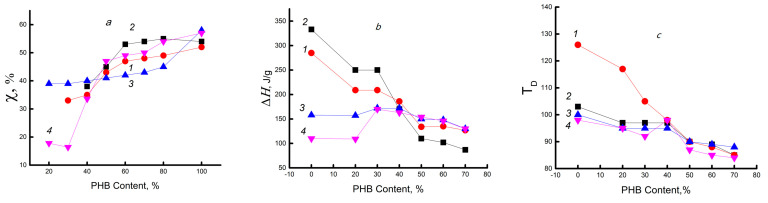
Dependence of the thermal characteristics on the composition of the PHB/chitosan composition: (**a**) the degree of crystallinity of PHB (χ), (**b**) the enthalpy of cleavage of hydrogen bonds of chitosan (ΔH), and (**c**) the maximum temperature of decomposition of hydrogen bonds of chitosan (T_D_). The numbers on the curves: 1—binary composition of PHB/chitosan; 2—the same, but with encapsulated DPD; 3—binary composition of PHB/chitosan after exposure in the aquatic environment at 70 °C; 4—the same, but with encapsulated DPD after exposure in the aquatic environment at 70 °C.

**Figure 3 polymers-15-02260-f003:**
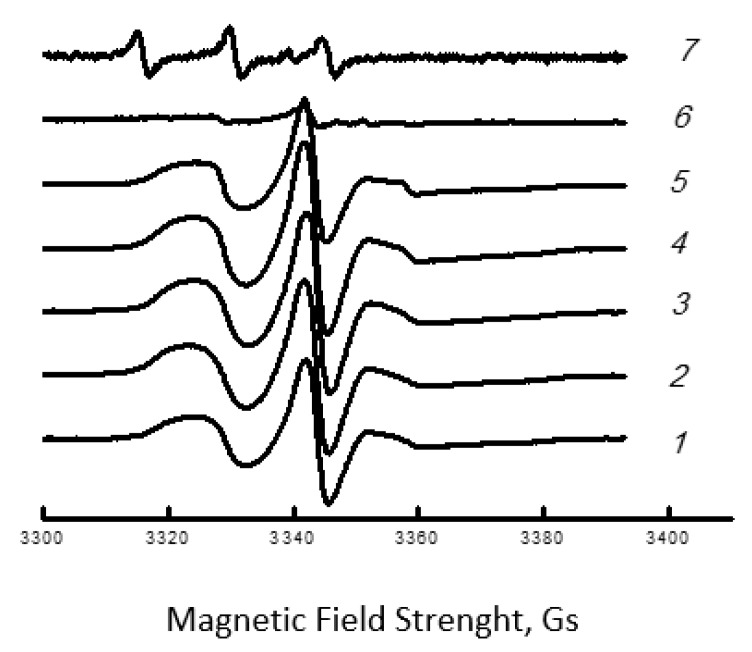
ESR spectra of nitroxyl radical TEMPO for the blends of PHB-chitosan with the content of the polysaccharide: 1—0, 2—30, 3—50, 4—60, 5—70, 6—80, 7—100%.

**Figure 4 polymers-15-02260-f004:**
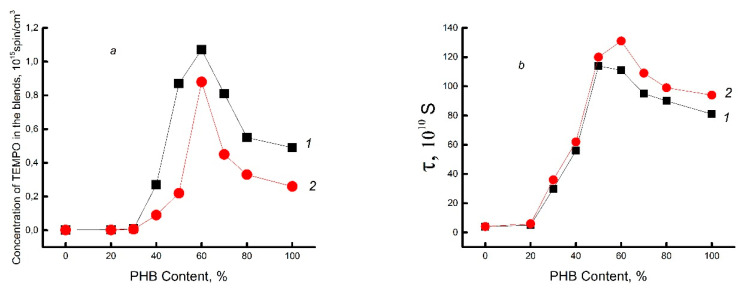
Concentration of TEMPO radical (**a**) and its time of correlation (**b**) in the PHB–chitosan blends. Binary system (1) and the same system with embedded DPD (2).

**Figure 5 polymers-15-02260-f005:**
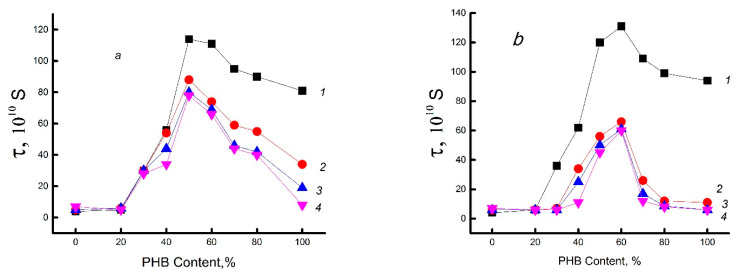
Characteristic time correlation of TEMPO as the blend characterization of hydrothermal action at 70 °C. Time of sample contact with the aqueous medium. (**a**) Initial binary system; (**b**) the same system with encapsulated DPD. Time of exposure: 1—0, 2—30, 3—60, 4—120 min.

**Figure 6 polymers-15-02260-f006:**
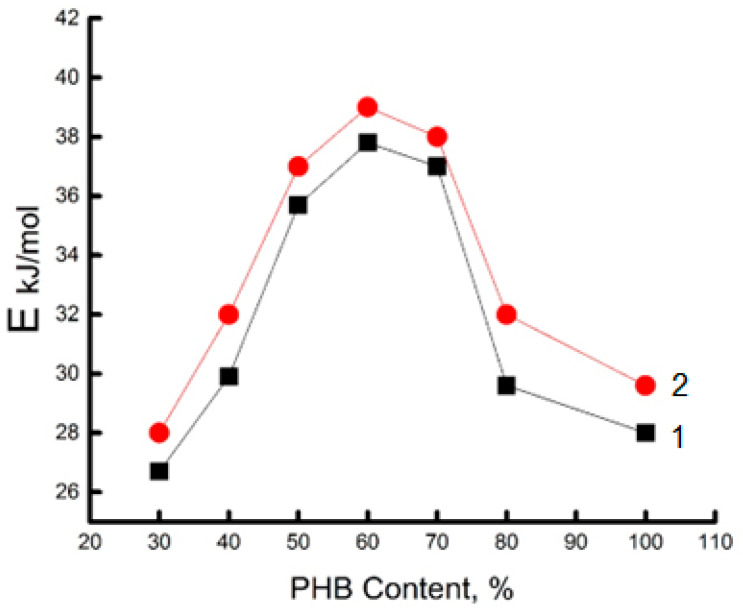
Effect of the composition of the PHB/chitosan mixture on the effective activation energy of the rotational mobility of the TEMPO molecular probe. 1—binary system, PHB-chitosan; 2—the same system with encapsulated DPD.

**Table 1 polymers-15-02260-t001:** Degree of crystallinity (χ), melting enthalpy (∆H, J/g), and melting point of PHB (T_m_, °C) in a mixture with chitosan, and the maximum temperature of chitosan hydrogen bond destruction (T_D_). All characteristics were obtained by DSC.

Initial Composition of PHB/Chitosan									
		PHB 100%	PHB/chit 80/20%	PHB/chit 70/30%	PHB/chit 50/50%	PHB/chit 40/60%	PHB/chit 30/70%	PHB/chit 20/80%	Chitosan 100%
PHB	χ, %	52	49	47	43	35	33	-	-
	T_m_	157;173	155;170	155;170	156;171	152;168	153	-	-
Chitosan	∆*H*	-	-	128	134	186	205	209	285
	T_D_	-	-	85	95	132	98	117	126
**PHB/chitosan with DPD**									
PHB	χ, %	54	55	54	45	38	-	63	-
	T_m_	153;170	153;168	153;168	151;167	151;167	-	163	-
Chitosan	∆*H*	-	-	87	110	101	248	250	333
	T_D_	-	-	85	90	97	96	97	103
**PHB/chitosan in the Aquatic Environment**									
PHB	χ, %	58	45	43	41	40	39	39	-
	T_m_	155;170	155;170	155;170	155;170	155;170	155;170	171	-
Chitosan	∆*H*	-	115	130	159	172	172	157	156
	T_D_	-	87	88	89	90	95	95	95
**PHB/chitosan with DPD, in the Aquatic Environment**									
PHB	χ, %	80	76	70	66	47	23	25	
	T_m_	153;170	152;169	152;169	152;168	152;168	172	172	-
Chitosan	∆*H*	-	-	130	154	163	170	109	110
	T_D_	-	-	85	87	97	92	95	95

## Data Availability

Not applicable.

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
