# Peer review of "Effect of Drug Encapsulation and Hydrothermal Exposure on the Structure and Molecular Dynamics of the Binary System Poly(3-hydroxybutyrate)-chitosan"

_polymers, 2023, doi:10.3390/polym15102260_

Round 1

Reviewer 1 Report

1. Rewrite the introduction and organize the backgrounds in a logical flow.

2. The method should be effectively described. The preparation of the sample. Describe the method of each parameter.

3. Conclusion shoild be summarized in one paragraph.

4. The reference should be updated such as 1996, 1999, 1987, etc..

Author Response

Dear Review! Thank you very much. 

The authors of given submission express the profound appreciation for the valuable and constructive correction and remarks that essentially improved the quality of the manuscript.

  • The introduction was dramatically rewritten; the logic of intention for the presentation was made considerably clearer to emphasize background and to make the literature survey more concise.
  • The method section has been expanded
  • Conclusion was significantly shortened and formulated as one paragraph.
  • In the list of citation, all the references older than 2000 have been updated. Currently, there are no any reference later this year. Besides, the authors have removed the self-citation item and replaced it by the published work of the other authors.

Reviewer 2 Report

polymers-2363097

The present manuscript deals with polymer films of poly-(3-hydroxybutyrate) [PHB] and chitosan with different compositions. Such films or materials can be used as drug delivery systems. Dipyridamole [DPD] is used as an example drug. By means of ESR spectroscopy and the stable spin probe TEMPO, the influence of the mixtures on amorphous and crystalline domains is investigated via the mobility of the probe. Using DSC, the H-bridge network of chitosan is investigated as well as the crystallinity of PHB.

Dear Authors,

The methodology described in the manuscript is a very interesting approach to study amorphous and crystalline domains in polymers.

Unfortunately, there are still some points that need clarification.

1) DSC measurements: Unfortunately, nothing is reported anywhere about the sensitivity of the samples to thermal treatment. Different thermal histories can significantly change the DSC data. It would be important to know, how sensitive the studied materials are in this regard.

2) Since the two polymers are incompatible, it would be important to know if and how the drug is distributed among the phases. How is it present amorphous or crystalline?

Material and Methods:

3) The conditions of water storage of the samples modified with active ingredient is not described.

4) Also not described here is how the active ingredient gets into the polymer and at what concentrations.

5) The active ingredient DPD is also crystalline. It has to be discussed if and how this influences the DSC measurements.

6) Line 118: Mw is given in KDa, this should then be the case for line 120 as well, or omit.

7) Line 121 replace "pouring" with film casting.

8) Line 122: What does ChDA mean?

9) Where does the spin probe come from? The process of how the probe gets into the polymer is not adequately described. Line 131 talks about fibers and not films!

10) Line 138: Equation 1 should not show the large spaces.

11) Line 146: The amount weighed into the sample pan in the DSC measurements is not specified. Line 503 and Acknowledgments report that a Netsch instrument was used for the DSC measurements. What was it actually then?

Results and Discussion

12) Line 178: The literature reference is strange. This also occurs in line 262.

13) Figure 2: Graph (a): Typo at a. Is also duplicated.

14) Fig 2 signature: (a) the degree..., Typo

15) How was the crystallinity (Chi) determined exactly. As compositions change, it is my understanding that the crystallinities of the total sample also change. This should be discussed. These measurements are very important for the statements in the paper, so this should be well understood.

16) Lines 256 to 259: Since DPD is also crystalline, it needs to be discussed how this affects the measurements. It is often observed that crystalline drugs are also present in polymer matrices in crystalline form. Uniform distribution is not the case for many active ingredients. Here, the process of how the drug-loaded films were prepared is very important. This problem is well known from the development of drug-eluting stents.

17) Line 259 talks about a plasticizing effect. Where exactly is this mentioned in the manuscript? Is there any literature on this? I have not found that.

18) From line 283. how much drug is released in the time of 2 hours. How much remains in the polymer. The crystallization tendency of the active ingredient must also be considered in this discussion.

19) Line 355: Figure 3 does not exist.

20) Signature Figure 4: .... Binary system (1) and the same system with embedded DPD (2)

21) Line 405: biopolyphyr, Typo

22) Line 439: Typo incapsulation.

23) Figure 6: Axis labeling and numbering of the graphs with 1 and 2.

24) Line 513: If there are IR data, it would be interesting to know if the statements about crystallinity can be supported with IR.

no

Author Response

Dear Review. Thank you very much.We removed the corrections from the manuscript. 

Reviewer 3 Report

Dear Authors,

Please read the comments I have included in the attached file.

Author Response

Dear Review. Thank you very much. The authors of given submission express the profound appreciation for the valuable and constructive correction and remarks that essentially improved the quality of the manuscript.

Thank you very much. We have corrected our manuscript.

Round 2

Reviewer 1 Report

It can be accepted